# Estimate Anaerobic Work Capacity and Critical Power with Constant-Power All-Out Test

**DOI:** 10.3390/jfmk9040202

**Published:** 2024-10-24

**Authors:** Ming-Chang Tsai, Scott Thomas, Marc Klimstra

**Affiliations:** 1Canadian Sport Institute Pacific, Victoria, BC V9E 2C5, Canada; mtsai@csipacific.ca; 2Graduate Department of Exercise Sciences, Faculty of Kinesiology and Physical Education, University of Toronto, Toronto, ON M5S 2W6, Canada; scott.thomas@utoronto.ca; 3School of Exercise Science, Physical and Health Education, University of Victoria, Victoria, BC V8W 2Y2, Canada

**Keywords:** training monitoring, fitness testing, endurance, performance

## Abstract

**Background:** The critical power model (CPM) is used extensively in sports to characterize fitness by estimating anaerobic work capacity (W’) and critical power (CP). Traditionally, estimates of CP and W’ require repeated, time-consuming tests. Alternatively, a 3 min all-out test yields good estimates of W’ and CP. However, adoption of the 3 min protocol for regular fitness monitoring is deterred by the mentally/physically strenuous nature of the test. **Objective:** We propose to examine an alternative single-session testing protocol that can accurately estimate critical power model parameters. **Methods:** Twenty-eight healthy competitive athletes (cyclists or triathletes) (mean ± SD: age: 38.5 ± 10.4 years, height: 177.9 ± 8.6 cm, mass: 73.4 ± 9.9 kg) participated in 5 sessions on a Lode cycle ergometer in isokinetic mode within a 2-week period. A 3 min all-out test (3MT) was conducted on the first visit to determine CPM parameters from which power outputs for 4 subsequent constant-power plus all-out tests (CPT) were selected to result in exhaustion in 1–10 min. The subjects were to maintain the prescribed power output as consistently as possible at their preferred race cadence. Once the power output could no longer be maintained for more than 10 s, the subjects were instructed to produce an all-out effort. Tests were terminated after power output fell to an asymptote which was sustained for 2 min. **Results:** The CPM parameters for all of the CPT durations were compared to the traditional CP protocol (significant parameter differences were identified for all CPT durations) and the 3MT (only CPT durations > 3 min were different [3–6 min test, *p* < 0.01; >6 min test, *p* < 0.01]). CPT does not estimate traditional CP and W’ parameters well. However, the CPT with a duration < 3 min accurately estimates both parameters of a 3MT. **Conclusion:** Therefore, CPT has the capacity to serve as an alternative tool to assess CP parameters.

## 1. Introduction

The ability to monitor fitness is critical for athletes and coaches. Several methods (such as $VO_{2max}$ [1] and blood lactate threshold tests [2]) used to assess fitness currently exist, but they have important shortcomings. These tests can be cumbersome ($VO_{2max}$ [3]), invasive (lactate threshold test [4]), and expensive, making them more suitable for high-performance and elite athletes. Additionally, these tests disrupt an athlete’s structured training schedule because they require the athlete to be well rested for accuracy [4]. Furthermore, while $VO_{2max}$ provides an upper limit of aerobic capacity, it does not capture the anaerobic capacities that are essential for performance in high-intensity events [3]. Similarly, blood lactate threshold tests are limited, as they often require extensive recovery and can be influenced by factors such as an athlete’s training state and nutrition [4]. Therefore, there is a need for a simple, more accessible technique to help athletes of all levels monitor their fitness status.

The well-established critical power model (CPM) is used to characterize an athlete’s work performance ability by estimating his/her anaerobic work capacity (W’) and critical power (CP). The aerobic component is called critical power (CP) and theoretically represents the power output that one could maintain indefinitely without reaching exhaustion [5]. Work capacity (W’) theoretically represents a finite supply of energy that is available for powering activity at intensities greater than CP and has been associated with anaerobic energy provision. By estimating both CP and W’, the CPM provides a more comprehensive view of an athlete’s performance potential, bridging the gap left by traditional tests like $VO_{2max}$ and lactate threshold assessments, which primarily focus on aerobic capacity and do not adequately assess anaerobic performance [6]. Total W’ depletion results in exhaustion, while partial depletion restricts the athlete’s maximal power output [6]. The CPM parameters can be used to predict performances [7] and recommend specific training intensities [8]. The model has been routinely applied to cycling [9] and gradually extended to other sports, including running [10], swimming [11], and rowing [12]. Traditionally, several constant-power exhaustive tests (3–5) are required to estimate these two parameters [13,14]. This impractical protocol can take upwards of 1–1.5 weeks to complete depending on the number of tests performed and the recovery periods between tests.

Previous studies found that anaerobic capacity could be predicted from one all-out exercise lasting 90–120 s [15,16]. To address the shortcomings of the time-consuming, multi-day protocol traditionally required for the CPM, a single 3 min all-out test (3MT) was proposed to estimate CPM parameters [17] with the rationale that once W’ is completely depleted after 90–120 s, the remaining maximum power output (termed end test power, EP) should equal the CP [17,18]. The parameter estimates from the 3MT were validated with the estimates derived from the traditional CP model [18,19], and the reliability analysis showed that repeated 3MT can provide reliable CP and W’ estimates [20,21]. One training intervention study has shown that 3MT can reflect the training-induced changes in CP [22]. This demonstration of sensitivity to change has a significant practical application in monitoring adaptation to training. Moreover, power duration-based training intensity zones have been created using 3MT end power to allow for constant monitoring of real-time performance [8].

Despite the appeal of the 3MT, the mentally and physically exhaustive nature of the test deters repeated testing occurrences. Thomas et al. (2012) reported that adopting an even pacing strategy (constant work-rate tests) reduces the perception of exertion compared to a self-paced (aggressively paced) strategy [23]. de Koning et al. (2011) theorized that perceived exertion (RPE) at any given time point is dependent on the magnitude and rate of homeostatic disturbance and the fraction of duration or distance remaining [24]. In other words, a fast start would result in higher reported RPE values for the entire race, resulting in an increased “hazard of catastrophic collapse” [24]. This drawback may be one of the factors that impairs our ability to use the 3MT as a regular fitness monitoring tool. Consequently, this approach to estimating anaerobic and aerobic fitness components is not widely employed. Therefore, there is a need for a less physically and mentally exhausting protocol to measure aerobic and anaerobic components.

Our purpose in this paper was to develop an alternate protocol which would produce useful estimates of aerobic and anaerobic fitness with decreased impact on the athlete. In comparison with the established test protocols, our constant-power plus all-out test (CPT) addresses the drawbacks of the impractical lengthy nature of the traditional CP test [25] and the mentally and physically exhausting nature of the 3MT [17].

## 2. Materials and Methods

Twenty-eight healthy competitive athletes (22 males and 6 females) who participated in either cycling or a triathlon (10 triathletes and 18 cyclists) volunteered to take part in this study (mean ± SD: age: 38.5 ± 10.4 years, height: 177.9 ± 8.6 cm, body mass: 73.4 ± 9.9 kg). The recruitment period lasted 2 months, and participants were recruited from the University of Toronto Masters Running Club and Triathlon Club in Toronto, Canada. All of the participants were accustomed to high-intensity exercise and had been involved in cycling for 3–20 years (9.38 ± 7.58 years). Prior to each session, study participants were requested to refrain from participating in strenuous physical activity in the preceding 24 h and to refrain from caffeine and alcohol for the 3 h before reporting to the laboratory. They were informed about the aims, the procedures, and the risks associated with the tests, and written informed consent was obtained by all participants. The study was approved by the University of Toronto Review Ethics Board and was conducted in accordance with the Declaration of Helsinki.

Participants were screened using the PAR-Q and athlete consent form in person. PAR-Q+ is a physical activity readiness questionnaire that is used to determine the safety and possible risks for an individual beginning an exercise program [26]. The questionnaire has been used in several studies that involve maximum efforts such as $VO_{2max}$ [27,28,29]; therefore, it is an acceptable screening process for the current study. Once the written consent was obtained, subjects were initially familiarized with all protocols and procedures. Each participant visited the laboratory on five occasions, with a minimum 48 h of recovery between each test, and all tests were completed within 14 days. Only one exercise test was conducted per day, with no more than three experimental sessions scheduled in any given week. The subjects exercised on a computer-controlled, electromagnetically braked cycle ergometer in isokinetic mode (Excalibur Sport, Lode, Groningen, the Netherlands) with the cadence fixed at the subject’s preferred racing cadence (range 85–105 rpm). Cadence is known to affect both end power in 3MT and CP [13]. Since power is a product of both torque (force applied by the muscles) and angular velocity (cadence), in order to isolate the power that is generated from the muscle, force cadence was fixed. Since the relationship between pedal cadence and ergometer load impacts EP [18], a fixed cadence was used to more accurately estimate CPM parameters.

During the first visit, an estimate of the subject’s critical power (CP) and anaerobic work capacity (W’) was determined using the three-minute protocol of Vanhatalo and colleagues [18]. In the subsequent four visits, the subjects were randomly assigned to four power outputs (calculated as power-1/time CPM) that resulted in exhaustion between 1–10 min followed immediately by a non-disclosed all-out test.

Visit 1: 3 min all-out test. Before each trial, subjects performed their regular race warm up protocol lasting between 10–20 min and then had 5–10 min of rest. The trial started with 1 min of easy cycling at <100 W. The subjects were asked to increase their effort five seconds before the commencement of the all-out effort. At the start of the effort and on the word “go”, the subject began sprinting maximally in a seated position at a constant cadence (i.e., isokinetics). The resistance of pedaling during the all-out effort was automatically adjusted by the Lode ergometer based on the subject’s pedaling effort in order to maintain cadence at the subject’s preferred race cadence. Subjects were instructed to reach their peak power as quickly as possible and to maintain an all-out effort for the entire test duration, thus avoiding pacing. Verbal encouragement was given throughout the tests, although no elapsed time or power feedback were given to the subjects during the test so as to avoid pacing. Subjects were instructed and strongly encouraged to provide maximum effort at all times throughout the test. The end power (EP) was determined as the mean power output during the final 30 s of the test, and the W’ was estimated as the power–time integral above the end power [18].

Visit 2–5: Constant-power plus all-out tests. Each subject completed a randomized series of four constant-power plus all-out tests (CPT) to exhaustion, with each test implemented at a different power output chosen to result in exhaustion between 1–10 min. These tests were performed after the subject’s preferred warm up. The subjects were asked to maintain the prescribed power output as consistently as possible at their preferred race cadence. The only feedback available to the subjects was the instantaneous power output in order to help maintain effort consistency. Once the subjects could no longer maintain the prescribed power output for more than 10 s, an all-out effort was instructed and power output feedback was removed until the termination of the test. Tests were terminated after power output fell to an asymptote and was sustained for one minute (see Figure 1) based on our own pilot work which demonstrated a reproducible leveling out of power output.

The traditional CP and W’ were calculated from the duration for which the constant power was maintained for the 4 constant-power tests. The EP was calculated as the average power output for the final 30 s of the test, and W’ was estimated as the power–time integral above the EP. The EP from the 3 min all-out test (3MT) was used as a reference EP for the 4 CPTs to determine the individual test W’s. Two W’s were extracted from each test, constant-power and unaccounted. The constant-power W’ was the region bounded by the duration spent maintaining constant power and EP, while unaccounted W’ was bounded by the region immediately after the constant-power region when power output gradually decreased to an asymptote EP (Figure 1). Finally, the total W’ was the sum of the constant-power and unaccounted W’s.

Preliminary observation showed several longer-duration (>5 min) test sessions occasionally resulted in subjects’ average power outputs being lower than their 3MT EP, and this was more specifically associated with subjects with a lower W’. Fluctuations of the power output from second to second may result in the average power output for long-duration tests being a few watts higher or lower than prescribed. If the prescribed power output is slightly higher than the CP, then it would be possible for the average EP to be lower than the 3MT EP, resulting in a negative W’ estimate. Therefore, an alternate approach was employed in calculating the constant-power W’s which used the EP for the individual test.

### Statistical Analysis

Comparisons of the EP, constant-power W’, unaccounted W’, and complete W’ between all of the tests were assessed using a one-way repeated-measures analysis of variance (ANOVA) mixed model in R (Version 4.4.1; Vienna, Austria) with compounded symmetry assumption made based on the variance–covariance parameter. The Dunnett–Hsu post hoc procedure with multiple comparison control was used to control for type I error in multiple comparisons in order to determine the significant difference between the parameters determined from all the tests and the parameters estimated from the CP model. A significance level (α) of 0.05 indicated that a difference was presented when no actual difference existed. Investigation of the residual plot showed a random scatter of points, and the normality plot showed that the residuals fell on a straight line, indicating that the normality assumption was appropriate for both CP and W’.

## 3. Results

The mean and standard deviation values of all the calculated variables for each test are reported in Table 1. One-way repeated-measures ANOVA revealed significant differences (*F*(5,158) = 2.083, *p* = 0.07) between the all-out test EPs and CP (Figure 2). Significant differences were detected in W’ between CPT and 3MT (*F*(5,158) = 15.2, *p* < 0.01), while post hoc analysis showed only for CPT durations > 3 min (3–6 min test *p* < 0.01, and >6 min test *p* < 0.01, respectively).

The constant-power W’ for all of the tests appeared to increase with test duration, while significant differences were only observed between test durations over 3 min and constant-power W’ (3–6 min *p* < 0.01, >6 min *p* < 0.01) (Figure 3A). On the other hand, the unaccounted W’ decreased as test duration increased. Significant differences were observed between unaccounted W’ for test duration less than 1 min and other test durations (1–3 min *p* < 0.01, 3–6 min *p* < 0.01, >6 min *p* < 0.01) (Figure 3B).

Total W’ (combining constant-power and unaccounted) for all the tests was higher than the traditional W’, primarily due to the addition of the unaccounted W’. However, only the W’ from the >6 min test showed significant difference from the 3MT test’s W’ (*p* < 0.01) (Figure 4).

Upon reanalyzing our data using the Bonferroni post hoc procedure, as was done in Dekerle’s paper, we found no statistically significant differences in 3MT EP (*p* = 0.06) and CPT EP for durations ≤ 3 min (for ≤1 min, *p* = 0.18, and for 1–3 min, *p* = 0.06) when compared to CP. This difference in statistical approach resulted in an incorrect rejection of the hypothesis, as the differences observed were only moderately significant.

## 4. Discussion

The results suggest that 3MT and CPT do not provide an accurate measure of the traditional CP and W’. More specifically, mean power outputs during the final 30 s are lower and W’ is higher for all the tests than estimates of CP and W’ derived from the traditional power-1/t model. However, the same estimates determined from CPT with durations ≤ 3 min are no different from those determined from the 3MT.

Our observation of lower 3MT and CPT EPs, using the multiple comparisons with control procedure, is inconsistent with the findings reported by Dekerle [30]. Dekerle compared 3MT EPs and CPs for two different cadences (60 rpm vs. 100 rpm) and applied the Bonferroni procedure to correct for Type I error when significance was detected in a two-way ANOVA with repeated measures. The Bonferroni procedure adjusts the *p*-value by dividing α by the number of comparisons (k), where k was 6 in Dekerle’s case, resulting in an adjusted *p*-value threshold of *p* < 0.01 (assuming α = 0.05). The inclusion of unnecessary comparisons (e.g., CP at 60 rpm vs. 3MT at 100 rpm and CP at 100 rpm vs. 3MT at 60 rpm) artificially lowers the *p*-value threshold, potentially leading to an incorrect rejection of hypotheses. Therefore, in our study, we used the multiple comparisons with control procedure [31] to reduce the risk of Type II error.

CP has been shown to be not significantly different from the EP obtained from constant-load 3 min all-out test [18]. However, in our isokinetic protocol, we observed a 4% lower power output when compared to CP. These two protocols differed in terms of control of power outputs, of the cadence in the constant-load mode, and of the force in the isokinetic mode. The end cadence in the constant-load 3MT as power output declines to a stable value is generally different from the subject’s preferred cadence. It has been shown that an end cadence at or slightly below the subject’s preferred cadence provides robust and accurate estimates of the CP model, but higher cadences reduce the CP [23]. Hence, this difference in end cadence may explain the decrease in isokinetic 3MT EP due to the power–velocity relationship of the muscles involved in cycling [32].

There was no difference in EPs between 3MT and CPT for durations ≤ 3 min, but significantly lower power outputs were observed for 3–6 min and >6 min (5.7% and 8.9%, respectively). Vanhatalo et al. found that iEMG during the 3MT progressively decreased [33], suggesting that all muscle fibers were activated from the onset, preventing further recruitment [34,35,36]. In contrast, during a 3 min work-matched CPT, iEMG increased, indicating progressive recruitment of higher-order (type II) fibers, with peak values at the limit of tolerance [33]. As these fibers fatigued, power output shifted to type I fibers [13,34,37]. Muscle fibers differ in fatigue resistance [38,39]: fast-twitch fatigable fibers (FF or type IIb) fatigue in <1 min, intermediate fibers ($F_{Int}$ or type IIx) in <5 min, and fatigue-resistant fibers (FR type IIa) and slow-twitch fatigue-resistance fibers (S or type I) can last up to 30 min or more. By the end of the 3MT, only FF fibers are fully fatigued, while some intermediate and all FR and S fibers remain active. Complete fatigue of FF and intermediate fibers would take over 5 min, at which point true CP is reached, predominantly generated by FR and S fibers. The lower power output observed beyond 6 min supports this interpretation; thus, we suggest treating this lowered power output as a true CP, which is the border between exercise intensities requiring aerobic-only and mixed aerobic and anaerobic energy production as defined by critical power concept.

Concerning estimates of W’, the data showed an increasing trend for the constant-power tests (Figure 3A), which is inconsistent with the traditional interpretation of W’ as a fixed anaerobic capacity [25]. For durations less than 3 min, the estimate for W’ is no different from traditional W’; however, there is a 266% and 364% increase in W’ for 3–6 min and >6 min test durations. This coincides with the duration at which EP significantly dropped lower than the traditional CP, hence resulting in an increase in W’. The physiological implications are likely to represent depletion in substrates (e.g., muscle phosphocreatine) for shorter and higher power output test durations and reuptake of metabolites (e.g., H+, inorganic phosphate, ADP, reactive oxygen species, and lactate) by neighboring oxidative muscle fibers at lower power-output, longer-duration tests [40,41]. In addition, W’ may not be completely depleted at the termination (i.e., inability to maintain constant power output) of constant-power tests [42]. If subjects were to continue after the termination of the test due to failure to maintain the selected power output, power output would gradually decline to an asymptote level (see Figure 1). Typically, this additional energy expenditure is not accounted for, which results in an underestimate of the complete W’. These unaccounted W’s were captured in our study, and a decreasing trend was observed as the test duration increased (Figure 3B). More specifically, the unaccounted W’ associated with the 1 min test was larger than all the other unaccounted W’s. At such a short duration, the constant-power W’ may represent the depletion of most if not all of the phosphagen (ATP-CP) system [43] and some anaerobic lactic system, while the unaccounted W’ represents the remaining glycolytic ATP production (anaerobic lactic system). As the test duration increases (>3 min), ATP-CP will occupy a smaller proportion of the constant-power W’, mostly being represented by the anaerobic lactic system, and the remaining anaerobic lactic system (approximately 20%) would be represented by the unaccounted W’ [44].

Combining the constant-power and unaccounted W’ provided us with the complete W’, which was shown to be larger than the traditional W’. However, constant-power exhaustive tests are prone to high variability [45], and given the multiple constant-power exhaustive tests required in the CP model, the compounded inherent variance would lead to potentially less-reliable calculation of W’ for the traditional method than W’ assessed using 3MT [20]. Comparisons of CPTs and 3MT showed no difference in the anaerobic capacity except for durations longer than 6 min. As mentioned earlier, reuptake of metabolites in the lower output for longer duration tests may contribute to the increase in capacity. An alternative explanation is the reciprocal relationship that was observed between the development of the $VO_{2}$ slow component and the progressive reduction in W’ [9,11,40]. Given that the $VO_{2}$ slow component is most pronounced in the lower region (higher intensity) of the severe domain [19,46], the respective W’ in the region would be lower in capacity. Similarly, if the output intensity is lower (still in the severe domain), then the $VO_{2}$ slow component would be smaller, and the subject would reach exhaustion after a longer period, resulting in higher W’. These results suggest that traditional interpretation of W’ as a fixed anaerobic work capacity may be outdated, as supported by work conducted by Vanhatalo in showing a decrease in W’ during exposure to hyperoxic gas [33] or an increase in W’ with priming performed exclusively in the heavy-intensity domain [47].

### Limitations

This study faces two main limitations that could affect the robustness and generalizability of the findings. First, the relatively small sample size of 28 participants, including 22 males and 6 females, which introduces an imbalance between sexes. This limited and unbalanced sample reduces the generalizability of the results, particularly when considering sex-based comparisons or extrapolating findings to the broader cycling population. The under representation of females may affect the robustness of any gender-specific conclusions. Future research should aim to include a more balanced and larger sample size in order to improve the reliability of the results and enhance generalizability across diverse populations. Second, the differences in power output protocols, particularly between isokinetic and constant-load 3 min all-out tests (3MT) present a limitation. The difference in cadence control—force-driven in isokinetic tests versus cadence-driven in constant-load tests—may influence the estimation of critical power (CP) and W’. This variation is further compounded by individual differences in preferred cadence, as it has been shown that estimates of CP can be influenced by end cadence, with higher cadences lowering CP values [48]. Hence, variations in muscle activation patterns and fiber recruitment across protocols may lead to inconsistent results, especially when comparing across different test durations and modes of power control. Consequently, these discrepancies could affect the accuracy and comparability of CP and W’ measurements between protocols. Further research is needed to investigate how such methodological differences impact the broader applicability of CP models.

## 5. Conclusions

The results from this study demonstrated that the constant-power all-out test addresses the drawbacks of time-consuming multi-day testing protocol in the traditional CP model as well as the mentally and physically exhaustive nature of 3 min all-out tests. Therefore, the constant-power all-out test has the capacity to serve as an alternative tool to assess CP parameters.

## Figures and Tables

**Figure 1 jfmk-09-00202-f001:**
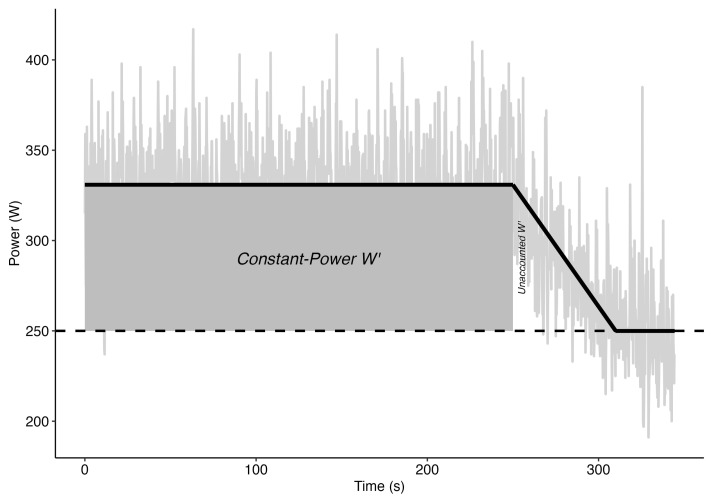
Sample 1 min test.

**Figure 2 jfmk-09-00202-f002:**
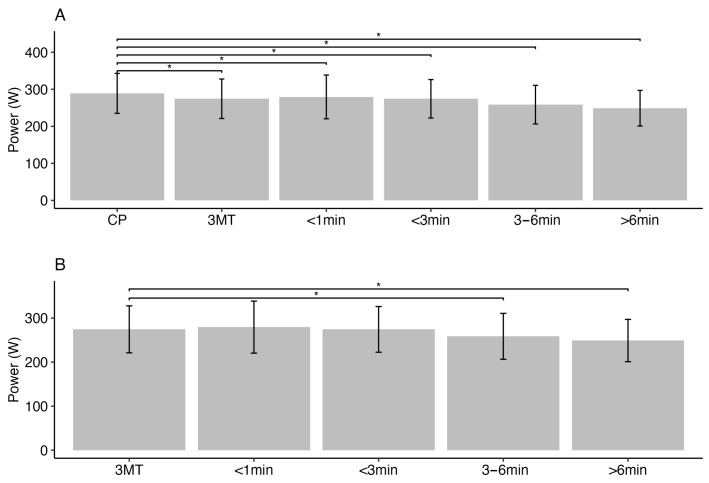
Comparison of end powers for different-duration constant-power tests to CP (**A**) and 3MT (**B**). (* *p* < 0.05).

**Figure 3 jfmk-09-00202-f003:**
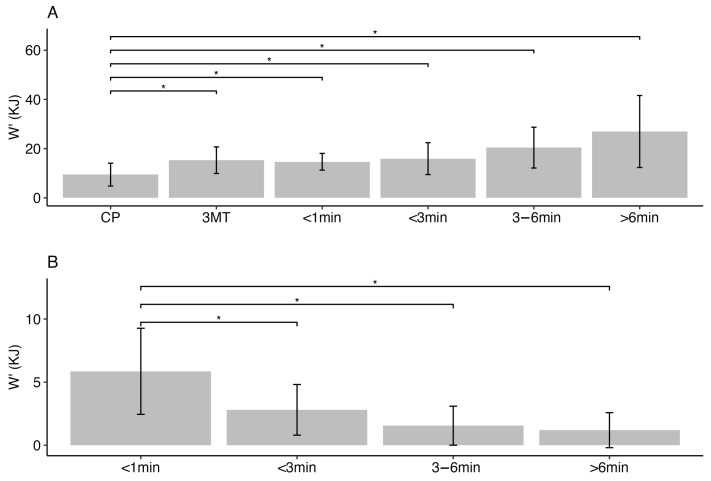
Constant-power W’ (**A**) and unaccounted W’ (**B**) for different test durations. (* *p* < 0.05).

**Figure 4 jfmk-09-00202-f004:**
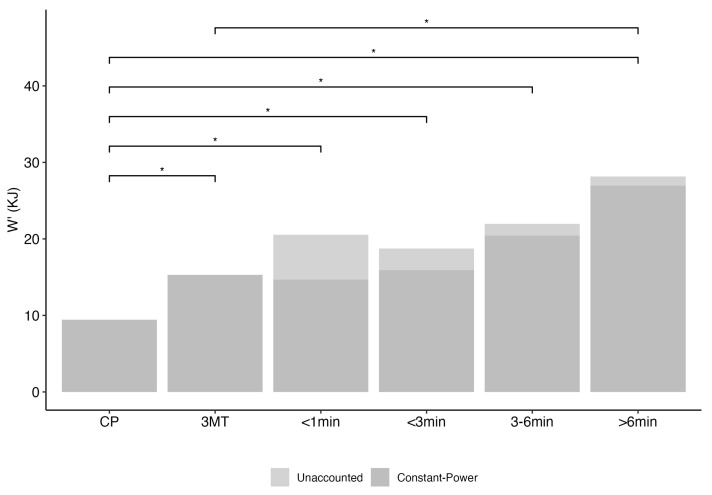
Total W’ for different test durations. (* *p* < 0.05).

**Table 1 jfmk-09-00202-t001:** CP or EP and W’s for different tests (mean ± SD).

	Traditional CP	3MT	CPT < 1 min	CPT 1–3 min	CPT 3–6 min	CPT > 6 min
CP or EP (W)	289 ± 54	274 ± 53	279 ± 59	274 ± 52	259 ± 52	249 ± 48
Constant-Power W’ (J)			8816 ± 3426	13,135 ± 5435	18,866 ± 7611	25,773 ± 13,955
Unaccounted W’ (J)			5859 ± 3409	2806 ± 2006	1550 ± 1545	1195 ± 1387
Total W’ (J)	9440 ± 4643	15,311 ± 5397	14,674 ± 3390	15,941 ± 6476	20,417 ± 8297	26,968 ± 14,624

## Data Availability

The original data presented in the study are openly available in GitHub at https://github.com/mingchangtsai/constantpower accessed on 15 October 2024.

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
