# Peer review of "Estimate Anaerobic Work Capacity and Critical Power with Constant-Power All-Out Test"

_jfmk, 2024, doi:10.3390/jfmk9040202_

Round 1
Reviewer 1 Report
Comments and Suggestions for Authors
Firstly, I would like to thank for inviting me to review this article. I found the work from Ming-Chang Tsai and colleagues important with several practical implications for coaches and sports practitioners. However, I have raised several important issues below that must be improved.
Major comments:
1. Based on the description, I assume this is a cross-sectional study. The Materials and Methods and Results are lacking some information. Please refer to the STROBE from EQUATOR and revise your article according to the unified guidelines. Please describe. All relevant issues are mentioned in those guidelines.
2. Competitive athletes with a mean age of 38.5 are quite too much for endurance and sprinting disciplines. Do you have exact data about participants’ training experience, number of competitions, etc?
3. Did you apply any selection protocol and evaluate inclusion criteria? If yes, describing them in the Materials and Methods is mandatory. Perhaps, a flowchart would be welcome. Did you assess their medical and health histories when including your athletes?
4. Where would you put your participants according to the McKay et al. classification system? https://pubmed.ncbi.nlm.nih.gov/34965513/
5. There is very limited information about your study group. Do you have more data about their demographics or training experience?
6. You applied an all-out test. How did you assess whether an athlete reached maximal effort? Did you use any kind of RPE scale (perhaps, the Borg scale)?
7. The description of statistical analysis is very limited. There is a lack of information about basic statistics used, and testing of assumptions (parametric distribution, etc.). There is no information on which P-value was considered significant.
8. Why have you used the Dunnet-Hsu post-hoc test for ANOVA? Other types of post-hoc tests are more widely used (e.g. Tukey post-hoc test).
9. You chose ANOVA as your leading statistical method. Moreover, you present only P-values in the text despite there should be other statistics presented along with P-values in ANOVA. Please see AMA 11thEdition on how to describe ANOVA results properly.
10. In lines 151-153, you describe it. There was no difference between sprinting and endurance sports. Could you provide data (perhaps in another table or the text) to support this claim?
11. In Results you only provide P-values. I recommend providing more details about your results (exact work etc.) directly in the text. However, remember to not repeat your tables.
12. Several places in the Discussion are lacking in references. For example, lines 237-240 would benefit from any support with reference.
13. The strong error in this study is a complete lack of description of limitations.
Minor comments:
1. I recommend adding a brief 1-2 sentence description of the study background in the abstract. Currently, the abstract begins with the aim.
2. Do not repeat the keywords of the terms used previously in the abstract. I suggest revising them to novel keywords.
3. Lines 21-25 are lacking in references.
4. In line 54 there should be a corrected reference bracket.
5. What was the recruitment period? Where (location) were the participants recruited? Please describe this issue in the Materials and Methods.
6. Please cite your ‘own pilot work’ in lines 119-121.
7. Please stratify your Materials and Methods for smaller paragraphs to facilitate reading the text.
8. I suggest rounding P-values to two decimal places.
9. In lines 170-173, you described the results of the Bonferroni correction applied to your data. I think it should be presented somewhere in the Results, not in the Discussion.
10. I think that the whole description of the Dekerle paper should be transferred to another paragraph. The first paragraph of the Discussion should present the primary study findings.
11. In my opinion, such a precise description of muscle fibers in the Discussion is not necessary and overloads the text. It is not the primary goal of this study.
Finally, to sum up my review, I congratulate the authors for their work. I strongly recommend considering my suggestion, especially the major points. Despite, several issues I raised, this paper is interesting for a broader audience.
Comments on the Quality of English LanguageThe English is generally correct. However, the text needs to be stratified into smaller paragraphs to facilitate its reading.
Author Response
Reviewer1
Firstly, I would like to thank for inviting me to review this article. I found the work from Ming-Chang Tsai and colleagues important with several practical implications for coaches and sports practitioners. However, I have raised several important issues below that must be improved.
Major comments:
- Based on the description, I assume this is a cross-sectional study. The Materials and Methods and Results are lacking some information. Please refer to the STROBE from EQUATOR and revise your article according to the unified guidelines. Please describe. All relevant issues are mentioned in those guidelines.
Response
The STROBE checklist does provide a useful set of items to consider. We have provided information which relates to the STROBE cross sectional study guideline as follows:
Title and Abstract. The Abstract now includes that this is a cross sectional study.
We employ a cross sectional study design to examine an alternative single session testing protocol
Methods
Study Design and Setting – key elements are provided including pre-test protocol, familiarization and testing time table. The laboratory setting is identified.
Participants. We provide this information in Responses to Comments 2, 3 and 5.
Variables and Measurements are carefully defined in the materials and methods. Measures based on the output from the Lode ergometer are detailed. For example: “The traditional and CP and W1 were calculated from the duration for which the constant power was maintained for the 4 constant power tests. The End Power was calculated as the average power output for the final 30s of the test and W’ estimated as the power-time integral above the EP.”
Bias. We describe how information to the participant was managed to reduce bias. The calculation of values was rigorously defined and automated to reduce experimenter bias.
Study Size. it is in line with on previous methodological studies such as those from Vanhatalo
Quantitative variables. These are described in the original Methods and Materials. Section 2.
Statistical Methods. These is dealt with in response to comments 7, 8 and 9.
Results and Discussion.
We have dealt with the particular issues raised in the comments below.
Please note that the STROBE guidelines were developed in the context of pathophysiology and epidemiology not in the context of developing an evaluation for sport science.
von Elm E, Altman DG, et al The Strengthening the Reporting of Observational Studies in Epidemiology (STROBE) Statement: guidelines for reporting observational studies. Ann Intern Med. 2007; 147(8):573-577. PMID: 17938396.
- Competitive athletes with a mean age of 38.5 are quite too much for endurance and sprinting disciplines. Do you have exact data about participants’ training experience, number of competitions, etc?
Response: training experience added as follows: “All the participants were accustomed to high-intensity exercise and had been involved in cycling for 3-20 years (sprint: 8.88 ± 6.90 years, endurance: 9.75 ± 8.25 years).”
- Did you apply any selection protocol and evaluate inclusion criteria? If yes, describing them in the Materials and Methods is mandatory. Perhaps, a flowchart would be welcome. Did you assess their medical and health histories when including your athletes?
Response: PAR-Q questionnaires was administered and added to the methods section as follows: Participants were screened using the PAR-Q and athlete consent form in person. PAR-Q+ is a physical activity readiness questionnaire that is used to determine safety and possible risks for an individual to begin exercise program (Thomas 1992). The questionnaire has been used in several studies that involve maximum efforts such as VO2max [Berry 2012, Silva-Cavalcante 2013, Hazell 2012), therefore it is an acceptable screening process for the current study.
- Where would you put your participants according to the McKay et al. classification system? https://pubmed.ncbi.nlm.nih.gov/34965513/
Response: 26 out of 28 participants would be classified as Tier 2 while the remaining 2 would be classified as Tier 3.
- There is very limited information about your study group. Do you have more data about their demographics or training experience?
Response: training experience and sport (cycling or triathlon) added as follows: Twenty eight healthy competitive athletes (22 males and 6 females) who participated in either cycling or triathlon (10 triathletes and 18 cyclists) volunteered to take part in this study (mean ± SD: age: 38.5 ± 10.4 years, height: 177.9 ± 8.6 cm, body mass: 73.4 85 ± 9.9 kg). The recruitment period lasted 2 months, and participants were recruited from the University of Toronto Masters Running Club and Triathlon Club in Toronto, Canada. All the participants were accustomed to high-intensity exercise and had been involved in cycling for 3-20 years (sprint: 8.88 ± 6.90 years, endurance: 9.75 ± 8.25 years).
- You applied an all-out test. How did you assess whether an athlete reached maximal effort? Did you use any kind of RPE scale (perhaps, the Borg scale)?
Response: RPE scale were not collected post session to validate maximal effort. However, participating athletes were aware that these were all out tests and verbal encouragement to provide maximum effort were given throughout the test. All participants were slumped over the handlebar motionless while gasping for air for few minutes after the end of the session could potentially indicate maximal efforts were given.
- The description of statistical analysis is very limited. There is a lack of information about basic statistics used, and testing of assumptions (parametric distribution, etc.). There is no information on which P-value was considered significant.
Response: Significance level (alpha) of 0.05 to detect significant difference, investigation of residual plots for constant variance, and Q-Q plots for normality statements added.
- Why have you used the Dunnet-Hsu post-hoc test for ANOVA? Other types of post-hoc tests are more widely used (e.g. Tukey post-hoc test).
Response: Tukey HSD is used to perform pairwise comparisons between all possible pairs of group means. Dunnett’s test is used to compare multiple groups to a single control group (ie. 3MT), not all pairwise comparisons. With the reduced number of comparisons, it will increase statistical power in relevant scenarios. Below is an example comparison of the p-value between the two statistical tests on CP/EP.
Comparisons |
Dunnett |
Tukey’s HSD |
3MT EP vs Traditional CP |
<0.01 |
0.02 |
CPT <1min vs Traditional CP |
0.05 |
0.13 |
CPT 1-3min vs Traditional CP |
<0.01 |
0.02 |
CPT 3-6min vs Traditional CP |
< 0.01 |
< 0.01 |
CPT >6min vs Traditional CP |
< 0.01 |
< 0.01 |
- You chose ANOVA as your leading statistical method. Moreover, you present only P-values in the text despite there should be other statistics presented along with P-values in ANOVA. Please see AMA 11thEdition on how to describe ANOVA results properly.
Response: ANOVA F-test added as per AMA guideline.
- In lines 151-153, you describe it. There was no difference between sprinting and endurance sports. Could you provide data (perhaps in another table or the text) to support this claim?
Response: sport type (sprinting or endurance) removed as it does not contribute to the focus of the paper.
- In Results you only provide P-values. I recommend providing more details about your results (exact work etc.) directly in the text. However, remember to not repeat your tables.
- Several places in the Discussion are lacking in references. For example, lines 237-240 would benefit from any support with reference.
Response: References added. Ettema 2009, Foss 2005
- The strong error in this study is a complete lack of description of limitations.
Response: Limitations to the study added in section 4.1 Starting with this statement:
. 4.1. Limitation 288
“This study faces two main limitations that could affect the robustness and generalizability of the findings.”
Minor comments:
- I recommend adding a brief 1-2 sentence description of the study background in the abstract. Currently, the abstract begins with the aim.
Response: study background added.
- Do not repeat the keywords of the terms used previously in the abstract. I suggest revising them to novel keywords.
Response: keywords changed to avoid duplicated words in abstract
- Lines 21-25 are lacking in references.
Response: References added.
- In line 54 there should be a corrected reference bracket.
Response: reference bracket added
- What was the recruitment period? Where (location) were the participants recruited? Please describe this issue in the Materials and Methods.
Response: Recruitment period and location added to the Materials and Methods section.
- Please cite your ‘own pilot work’ in lines 119-121.
Response: Our unpublished work cited.
- Please stratify your Materials and Methods for smaller paragraphs to facilitate reading the text.
Response: Methods section was split into smaller paragraphs for better readability
- I suggest rounding P-values to two decimal places.
Response: P-values rounded to 2 decimal places
- In lines 170-173, you described the results of the Bonferroni correction applied to your data. I think it should be presented somewhere in the Results, not in the Discussion.
Response: Bonferroni correction results moved to the Results section as suggested.
- I think that the whole description of the Dekerle paper should be transferred to another paragraph. The first paragraph of the Discussion should present the primary study findings.
Response: Dekerle paper discussion moved to a separate paragraph
- In my opinion, such a precise description of muscle fibers in the Discussion is not necessary and overloads the text. It is not the primary goal of this study.
Response: The paragraph on muscle fibers explanation of drop in EP with longer durations has been rewritten to be more concise.
Finally, to sum up my review, I congratulate the authors for their work. I strongly recommend considering my suggestion, especially the major points. Despite, several issues I raised, this paper is interesting for a broader audience.
Reviewer 2 Report
Comments and Suggestions for Authors
Basic reporting
Dear authors, the manuscript is generally well-written and easy to read; a slight spell-check is required. I have just some concerns that the authors must address.
Abstract
I suggest adding a brief background.
What’s the gap you’re trying to fill with your work?
keywords usually should be different from that used in the main title.
Introduction
The literature on the subject is sufficiently well summarised. However, it could be useful to add some information about:
- some key terms are not clearly defined. I.e., Critical Power Model (CPM) and W' are introduced without sufficient background for readers who might be unfamiliar with these concepts.
- You mentioned that VO2max and blood lactate threshold tests have shortcomings, but how CPM addresses those shortcomings it’s not fully explained. It mentions CPM being used to estimate anaerobic work capacity (W') and critical power (CP), but the connection between these and VO2max or lactate threshold is not immediately apparent.
- I could be wrong but: you state that 3MT is a less time-consuming alternative to traditional methods, but it is later criticized for being mentally and physically exhausting, which seems contradictory. If the test is so exhaustive that athletes are deterred from using it regularly, its practical value is diminished. This contradicts the initial claim that 3MT is a simple solution.
Methods
- Cadence is fixed at the subject’s preferred racing cadence (85-105 rpm) during the isokinetic testing. Can differences in cadence affect performance parameters like power output and fatigue rates?
- The sample size of 28 participants (22 males, 6 females) is relatively small, particularly with the imbalance between sex. The limited sample size reduces the generalizability of the results, especially when making gender-based comparisons or extrapolating to the broader cycling population. At least you should state it as a limitation.
Validity of the findings
- The logic around CP and EP differences seems unclear and contradictory. You state no difference for certain test durations, then mention significant differences at other points, without adequately explaining the conditions under which these differences occur.
- The physiological reasoning (muscle fiber recruitment and fatigue patterns) isn't supported by direct measurement in this study. You’re making assumptions about the underlying mechanisms without corresponding data, which weakens the argument.
- The recommendation of CPT as a superior alternative seems weak because yourselves admit inconsistencies in results. More justification is needed to support this conclusion, particularly given the contradicting findings regarding CP, EP, and W' across tests.
Comments on the Quality of English LanguageMinor editing of English language required.
Author Response
Reviewer2
Dear authors, the manuscript is generally well-written and easy to read; a slight spell-check is required. I have just some concerns that the authors must address.
Abstract
I suggest adding a brief background.
Response: study background added.
What’s the gap you’re trying to fill with your work?
We have addressed this in the introduction.
keywords usually should be different from that used in the main title.
Response: keywords changed to avoid duplicated words in the title
Introduction
The literature on the subject is sufficiently well summarised. However, it could be useful to add some information about:
- some key terms are not clearly defined. I.e., Critical Power Model (CPM) and W' are introduced without sufficient background for readers who might be unfamiliar with these concepts.
Response: CPM parameters definition added
- You mentioned that VO2max and blood lactate threshold tests have shortcomings, but how CPM addresses those shortcomings it’s not fully explained. It mentions CPM being used to estimate anaerobic work capacity (W') and critical power (CP), but the connection between these and VO2max or lactate threshold is not immediately apparent.
Response: statements on CPM addressing some of the shortcomings of VO2max and lactate threshold testing added.
- I could be wrong but: you state that 3MT is a less time-consuming alternative to traditional methods, but it is later criticized for being mentally and physically exhausting, which seems contradictory. If the test is so exhaustive that athletes are deterred from using it regularly, its practical value is diminished. This contradicts the initial claim that 3MT is a simple solution.
Response: 3MT was proposed as a simple alternative to the traditional CP assessment. However, its implementation has been limited due to the physically demanding nature of the protocol, which poses challenges for regular adoption in practice. The approach in this research was to try to further reduce the demands.
Methods
- Cadence is fixed at the subject’s preferred racing cadence (85-105 rpm) during the isokinetic testing. Can differences in cadence affect performance parameters like power output and fatigue rates?
Response: Thank you for your insightful comment regarding the effects of cadence on performance parameter. Cadence can significantly impact power output and fatigue rate in cycling. For instance, studies have shown that variations in pedaling cadence can affect cycling economy and efficiency, ultimately influence an athlete’s performance (Ettema 2009, Foss 2005). Additionally, it has been observed that cyclists may experience changes in their optimal cadence as workload increases, further emphasizing the relationship between cadence and performance outcomes (Ettema 2009, Foss 2005). These findings suggest that maintaining a fixed cadence, as employed in our protocol, is crucial for accurately assessing performance parameters during isokinetic testing.
- The sample size of 28 participants (22 males, 6 females) is relatively small, particularly with the imbalance between sex. The limited sample size reduces the generalizability of the results, especially when making gender-based comparisons or extrapolating to the broader cycling population. At least you should state it as a limitation.
Response: Limitation to the study added.
Validity of the findings
- The logic around CP and EP differences seems unclear and contradictory. You state no difference for certain test durations, then mention significant differences at other points, without adequately explaining the conditions under which these differences occur.
Response: Thank you for raising the issue of confusion around CP and EP. CP is the aerobic component derived from the traditional critical power model and EP is the aerobic component calculated from all-out tests. EP was validated by Vanhatalo and colleagues (Vanhatalo et al 2007). These terms were defined in the introduction and methods sections. In the first paragraph of the Results section stated “One-Way repeated-measures ANOVA revealed significant differences (F(5,158)=2.083, p=0.07) between the all-out test EPs and CP.”
- The physiological reasoning (muscle fiber recruitment and fatigue patterns) isn't supported by direct measurement in this study. You’re making assumptions about the underlying mechanisms without corresponding data, which weakens the argument.
Response: We agree that we did not make measures to substantiate these discussion points. We will identify that these are possible directions for future studies as follows: “Thus, we suggest treating this lowered power output as a true CP which is the border between exercise intensities requiring aerobic only and mixed aerobic and anaerobic energy production as defined by critical power concept. Further testing of the effect of all out tests on recruitment and fatigue patterns is required to support this interpretation.
- The recommendation of CPT as a superior alternative seems weak because yourselves admit inconsistencies in results. More justification is needed to support this conclusion, particularly given the contradicting findings regarding CP, EP, and W' across tests.
Response: We suggested CPT has capacity to serve as an alternative (not superior) assessment tool since <3min CPT EP is no different from 3MT EP. However, the CPT EPs (regardless of durations) were stated as significantly different from traditional CP, and this could be the possible misunderstanding of “inconsistencies” to which the reviewer refers.
Round 2
Reviewer 1 Report
Comments and Suggestions for Authors
The authors made necessary revisions and reply properly to my questions. I do not have any doubts or further comments.
Comments on the Quality of English LanguageThe English is generally fine.
Reviewer 2 Report
Comments and Suggestions for Authors
the author's adressed all my concerns. I have no further suggestions.